

# Association between fetal sex and metabolic syndrome in women aged 40 years and older: the REACTION study

Qian Xie[1,2], Ruoqing Li[3], Qin Wan[4] and Nanwei Tong[2]

[1] Department of Gerontology, the people's hospital of LeShan, Leshan, China
[2] Division of Endocrinology and Metabolism, West China Hospital, Sichuan University, Chengdu, China
[3] Department of General Medicine, Chongqing University of Medical Science, Chongqing, China
[4] Division of Endocrinology and Metabolism, The Affiliated Hospital of Southwest Medical University, Luzhou, China

Corresponding author
Nanwei Tong, tongnw@scu.edu.cn

## ABSTRACT

**Introduction:** This study aimed to investigate whether fetal sex influences the risk of metabolic syndrome (MetS) in women in China.

**Materials and Methods:** A cohort of 3,758 Chinese women, each having given birth to only one child and aged 40 years or older, was included in the study. Registry data on all pregnancies and miscarriages were collected for each participant. This retrospective cohort study aimed to determine whether fetal sex was associated with metabolic syndrome. MetS was defined as meeting at least three of the following five criteria: impaired insulin metabolism, reduced glucose tolerance, hypertension, dyslipidemia, and large waist circumference.

**Results:** Among the 3,758 women aged 40 years and older in Luzhou City, 1,018 (27.1%) developed MetS. Mothers who had carried a male fetus had higher triglyceride (TG) and total cholesterol (TC) levels and a greater prevalence of diabetes than those who had carried a female fetus ($P < 0.05$). Although the incidence of MetS was higher in women who had carried a male fetus than in those who had carried a female fetus, the difference was not statistically significant ($P > 0.05$). Notably, MetS was significantly less prevalent in premenopausal women than in postmenopausal women, irrespective of fetal sex ($P < 0.05$). In the postmenopausal subgroup, the incidence of MetS was significantly higher in women who had carried a male fetus compared to those who had carried a female fetus ($P < 0.05$).

**Conclusions:** Our study found no significant association between fetal sex and MetS in Chinese women aged 40 years and older. However, among postmenopausal women, the incidence of MetS was significantly higher in those who had carried a male fetus. Greater attention should be given to postmenopausal women with a history of carrying a male fetus, and early preventive measures should be implemented to reduce the risk of related chronic diseases.

## INTRODUCTION

Metabolic syndrome (MetS) is a cluster of interrelated conditions, including hypertension, hypertriglyceridemia with low high-density lipoprotein (HDL) cholesterol, insulin resistance, and large waist circumference (*Neeland et al., 2024*). Numerous evidence indicated that metabolic syndrome increases the risk of other diseases, such as thyroid disorders (*Jakubiak et al., 2024*), diabetes, cardiovascular diseases, cancer and premature death (*Rochlani et al., 2017*; *O'Neill & O'Driscoll, 2015*; *Ahmadinezhad et al., 2022*). MetS has been recognized as a global health concern with a rising prevalence, making it an emerging epidemic (*Saklayen, 2018*).

The prevalence of metabolic syndrome (MetS) increases with age, with women being more susceptible than men (*Pucci et al., 2017*). A study of 8,183 adults in the United States estimated that, based on the 2005 Adult Treatment Panel III (ATP III) criteria, the prevalence of MetS increased from 37.6% in 2011–2012 to 41.8% in 2017–2018 (*Liang et al., 2023*). This study also demonstrated a continuous increase in the occurrence of MetS. Additionally, based on the 2005 ATP III criteria, MetS was evaluated in 69,094 adults without cardiovascular disease (CVD) from the MORGAM Prospective Cohort Project, with a prevalence of 19.9% in men and 32.1% in women (*Vishram et al., 2014*). A review of several independent studies conducted in China found that 24.5% of the evaluated individuals had MetS, with a prevalence of 19.25% in men and 27.0% in women (*Li et al., 2016*). A study in Mexico involving 6,567 adults found that 28.9% of men and 44.4% of women are affected by MetS, a notably higher prevalence in women than in men (*Pérez-Castro et al., 2022*). According to a meta-analysis, the cardiovascular (CV) risk associated with MetS has been shown to be higher in women than in men (*Santilli et al., 2017*).

Multiple factors are believed to contribute to an increased risk of MetS. Live births, age at first pregnancy, age at last birth, and number of pregnancies play a crucial role in women's health. NHANES indicate that the association between age at first pregnancy and the risk of MetS follows an N-shaped curve in women. Additionally, a higher number of live births is associated with an increased incidence of MetS (*Zuo et al., 2023*). Fetal sex can influence intrauterine development and maternal health during pregnancy, while also having a substantial impact on the mother's long-term health and well-being after childbirth. Women carrying a male fetus have been shown to be at a higher risk of severe pregnancy complications and perinatal mortality (*Melamed, Yogev & Glezerman, 2010*). A recent study indicates that carrying a male fetus is associated with impaired maternal-cell function (*Retnakaran et al., 2015*). However, another study indicates that women carrying a female fetus (compared to those carrying a male fetus) have a higher risk of early postpartum progression to Type 2 diabetes (*Retnakaran & Shah, 2015*). Increased blood glucose is a component of metabolic syndrome. Limited research has been conducted on the relationship between fetal sex and MetS among Chinese women. This study aimed to investigate the association between fetal sex and the risk of MetS in Chinese women aged 40 years and older. Portions of this text were previously published as part of a preprint (https://www.researchsquare.com/article/rs-1833257/v1).

## METHODS

### Study population

This cross-sectional study was derived from the REACTION study (*Ning & Reaction Study Group, 2012*), a longitudinal study in China investigating the risk of cancer in Individuals with diabetes, which was performed between April and November 2011 by the Chinese Medical Association endocrine branch. Participants aged 40 and older were enrolled from five Luzhou communities using a multistage cluster random sampling method. All investigators were substantially trained to administer the study questionnaire and evaluate the outcome measures before starting the study, which received approval from the institutional review boards of the Ruijing Hospital Ethics Committee. In our study, the baseline data of 3,758 women aged 40 and older and each having given birth to a single child were selected. The inclusion criteria were as follows: (1) permanent residents aged ≥40 years; (2) gender: female; and (3) good compliance. The women were excluded for the following reasons: lack of mobility, advanced age (>85 years), weakness, communication barriers, poor compliance, a history of long-term chronic diseases and recently having suffered from acute disease (*Xie et al., 2022*).

The National Cholesterol Education Program (NCEP ATP III) criteria (*Alexander et al., 2003*) for MetS were used to classify the subjects without previously known MetS on the basis of three of the following five components: (1) abdominal obesity: male waist circumference ≥90 cm, female ≥85 cm; (2) increased blood glucose: fasting blood glucose ≥6.1 mmol/L, blood glucose ≥7.8 mmol/L 2 h after meal and/or diagnosed with diabetes, currently under anti-diabetes treatment; (3) hypertension: blood pressure ≥130/85 mmHg and/or diagnosed with hypertension; (4) fasting HDL-C <1.04 mmol/l; and (5) fasting triglycerides (TG) ≥1.7 mmol/L.

### Other measures

During the baseline interview, all participants were asked about sex, age, if diabetes in the family, level of education, age at menopause, miscarriage, age at which the first fetus was born, drinking and smoking habits, and the biological reproductive number, excluding other fetuses (adopted, foster, step). Smoking status was defined as currently smoking and smoking for 1 year prior to baseline.

During physical examination, blood pressure was measured with the individual seated, and measurements were performed three times and after resting for 5 min each time. Height and weight were obtained while wearing no shoes and light clothes. Body mass index (BMI) was calculated as weight (kg)/height (m)$^2$. Waist circumference was determined at the midway level between the costal margin and iliac crest.

Blood was drawn early in the day after fasting 8 h. All subjects underwent a glucose tolerance test (82.5 g of glucose monohydrate). Plasma glucose and hemoglobin A1c, triglycerides (TG), high-density lipoprotein cholesterol (HDL-C) and low-density lipoprotein cholesterol (LDL-C) were determined using the glucose oxidase assay and colorimetric enzyme assays. All study subjects gave informed consent.

## Statistical analysis

The Epidata program was used to establish the database. In order to verify whether the variable distributions conform to a normal distribution, we employed the Shapiro-Wilk test for evaluation. Continuous variables that conform to a normal distribution are expressed as the mean ± SD. Continuous variables that conform to a non-normal are expressed as the median. All reported $P$ values were two-sided, and <0.05 indicated differences with statistical significance. Comparisons between groups for normally distributed parameters were performed using ANOVA, while non-parametric variables were analyzed using the Kruskal-Wallis test. Discrete variables were compared using the Chi-square test, whereas continuous variables were compared using Student's t-test or the non-parametric Mann-Whitney test, depending on the normality of the variables. The association between fetal sex and MetS was evaluated using logistic regression. Statistical analyses were performed using SPSS version 19.0.

## RESULT

### Characteristics of the study participants according to fetal sex

A total of 3,758 cases of women, each having given birth to only one child and aged 40 years or older were selected to be studied. The mean age of the participants was 53.7 ± 8 years. We grouped them according to fetal sex: the mothers with male fetus group (2018, 53.70%), the mothers with female fetus group (1,740, 46.30%). The characteristics of each group are presented in Table 1. The mothers with male fetus group had higher TG and TC and greater background of diabetes than those with female fetus group ($P < 0.05$). The mothers with female fetus group had a lower age, FBG, 2hPG, HbA1c and TC than those with male fetus group, but without statistical significance ($P > 0.05$). The groups did not differ significantly in HDL-C, LDL-C, BMI, waist circumference, hip circumference, DBP, SBP, age at first delivery and menopausal status (Table 1).

### Comparison of the prevalence of MetS between the two groups

Among the 1,018 (27.1%) participants with incident MetS, MetS occurred more often in mothers with male fetus group but the difference was not statistically significant ($P > 0.05$) (Table 2).

### The distribution of MetS morbidity under different menstruation situations

Table 3 presents the distribution of MetS morbidity under different menstruation situations. MetS prevalence was lower in premenopausal women (15.7%) than postmenopausal women (31.6%), where there was a statistical difference in both the mothers with male fetus group and mothers with female fetus group ($P < 0.001$). The differences observed were not significant in premenopausal women between the mothers with male fetus group and the mothers with female fetus group ($P = 0.123$). However, the frequency of MetS was statistically greater in the mothers with male fetus group among postmenopausal women ($P < 0.05$) (Table 3).

**Table 1 Characteristics of the study participants according to fetal sex.**

| Variable | Mothers with a male fetus $n = 2,018$ | Mothers with a female fetus $n = 1,740$ | $t$ | $P$ |
|---|---|---|---|---|
| Age (years) | 54.0 ± 8.0 | 53.5 ± 7.8 | 1.699 | 0.089 |
| FBG (mmol/L) | 5.72 ± 1.49 | 5.69 ± 1.39 | 0.695 | 0.487 |
| 2hPG (mmol/L) | 7.33 (6.10, 9.35) | 7.17 (6.08, 8.97) | 1.434 | 0.152 |
| HbA1c (%) | 6.01 ± 0.96 | 5.98 ± 0.96 | 1.111 | 0.267 |
| TC (mmol/L) | 4.75 ± 1.14 | 4.64 ± 1.14 | 2.213 | 0.027 |
| TG (mmol/L) | 1.27 (0.90, 1.84) | 1.23 (0.87, 1.75) | 1.970 | 0.049 |
| HDL-C (mmol/L) | 1.31 ± 0.35 | 1.31 ± 0.36 | 0.225 | 0.822 |
| LDL-C (mmol/L) | 2.63 ± 0.82 | 2.58 ± 0.81 | 1.919 | 0.055 |
| BMI (kg/m$^2$) | 23.6 ± 3.2 | 23.5 ± 3.2 | 0.696 | 0.486 |
| Waist circumference (cm) | 80.0 ± 10.0 | 79.9 ± 9.7 | 0.273 | 0.785 |
| Hip circumference (cm) | 92.7 ± 9.5 | 92.7 ± 8.8 | 0.107 | 0.905 |
| SBP (mmHg) | 120.4 ± 18.7 | 119.6 ± 18.3 | 1.188 | 0.235 |
| DBP (mmHg) | 75.4 ± 10.6 | 75.2 ± 10.2 | 0.776 | 0.438 |
| Family history of diabetes | 23.1% | 14.3% | 110.905 | <0.001 |
| Age at first delivery | 20.6 ± 4.4 | 20.3 ± 4.8 | 0.746 | 0.474 |
| Menopausal status (Yes/No) | 1,465/553 | 1,226/514 | 2.115 | 0.146 |

Note:
Abbreviations: BMI, body mass index; WC, Waist circumference; HDL-C, high-density lipoprotein cholesterol; LDL-C, low-density lipoprotein cholesterol; TG, Triglyceride; TC, Total Cholesterol; SBP, Systolic Blood Pressure; DBP, Diastolic Blood Pressure.

**Table 2 Comparison of the prevalence of MetS between the two groups.**

| | MetS | Total | Morbidity (%) | $\chi^2$ | $P$ |
|---|---|---|---|---|---|
| Mothers with a male fetus | 571 | 2,018 | 28.3 | 2.751 | 0.097 |
| Mothers with a female fetus | 447 | 1,740 | 25.7 | | |

**Table 3 The distribution of MetS morbidity under different menstruation situations.**

| | Premenopausal MetS | Total | Morbidity % | Postmenopausal MetS | Total | Morbidity% | $\chi^2$ | $P$ |
|---|---|---|---|---|---|---|---|---|
| Mothers with a male fetus | 77 | 553 | 13.9 | 494 | 1,465 | 33.7 | 61.698 | <0.001 |
| Mothers with a female fetus | 91 | 514 | 17.7 | 356 | 1,226 | 29.0 | 20.064 | <0.001 |
| $\chi^2$ | 2.382 | | | 5.277 | | | – | – |
| $P$ | 0.123 | | | 0.022 | | | – | – |

## Comparison of the number of pregnancies and abortions

In Tables 4 and 5, we compared the number of pregnancies and abortion in women with and without metabolic syndrome with different fetal sex. Findings demonstrate that the mothers with female fetus group in women with metabolic syndrome had a higher number of pregnancies than the mothers with male fetus group, but without statistical significance ($P > 0.05$). The differences observed were not significant in the number of abortions

**Table 4 Comparison of the number of pregnancies.**

|  | MetS (+) | MetS (−) | Z | P |
|---|---|---|---|---|
| Mothers with a male fetus | 1.5 (1.0, 3.0) | 2.0 (1.0, 2.0) | 0.270 | 0.787 |
| Mothers with a female fetus | 2.0 (1.0, 2.0) | 1.0 (1.0, 2.0) | 0.540 | 0.589 |
| Z | 0.112 | 0.622 | – | – |
| P | 0.910 | 0.534 | – | – |

**Table 5 Comparison of the number of abortion.**

|  | MetS (+) | MetS (−) | Z | P |
|---|---|---|---|---|
| Mothers with a male fetus | 2.0 (1.0, 3.0) | 2.0 (1.0, 3.0) | 0.830 | 0.406 |
| Mothers with a female fetus | 2.0 (1.0, 3.0) | 2.0 (1.0, 3.0) | 0.712 | 0.477 |
| Z | 0.064 | 0.220 | – | – |
| P | 0.969 | 0.826 | – | – |

**Table 6 Multiple logistic regression analysis of risk factors for MetS.**

| Variable | B | S.E | Wald | P | OR | 95% CI* |
|---|---|---|---|---|---|---|
| Fetal sex | −0.018 | 0.306 | 0.003 | 0.953 | 0.982 | [0.539~1.789] |
| Menopausal status | −4.511 | 1.191 | 14.345 | 0.000 | 2.323 | [2.115–3.910] |
| The number of abortion | −0.081 | 0.451 | 0.032 | 0.857 | 0.922 | [0.381~2.233] |
| The number of pregnancies | −0.184 | 0.246 | 0.558 | 0.455 | 0.832 | [0.513~1.348] |

between the mothers with male fetus group and the mothers with female fetus group ($P > 0.05$).

## Multiple logistic regression analysis of risk factors for MetS

In the binary logistic regression model, we found that there is no significant association between fetal sex and MetS after adjusting for confounding factors ($P > 0.05$). We found that menopausal status [OR = 2.323 (2.115–3.910), $P < 0.001$] was statistically associated with an increased risk of MetS, whereas the number of pregnancies and the number of abortions were not. The analysis results indicated that fetal sex [OR = 0.982 (0.539–1.789), $P = 0.953$] was only a suspect factor, falling shy of statistical significance. Moreover, menopausal status was an independent risk factor for MetS following adjustment for other factors (Table 6).

## DISCUSSION

In the present study, we evaluated the association of fetal sex and MetS in Chinese women 40 and older. We found that the prevalence of MetS in women was 27.09%, which is lower thanpreviously reported data indicating a pooled incidence of MetS in China is 32.3% (29.2% in men and 35.4% in women) (*Huang et al., 2022*). Additionally, our findings revealed that the prevalence of MetS in premenopausal women was 15.7%, whereas it was

31.6% in postmenopausal women, aligning with results from a meta-analysis demonstrating a global prevalence rate of MetS in postmenopausal women of 37.17%, significantly higher than that observed in premenopausal women (*Cho et al., 2008*). At present, the pathogenesis of MetS remains unclear, and its occurrence is the result of a combination of factors including those of genetic and social environmental origin.

Our investigation indicated that mothers carrying female fetuses had lower fasting blood glucose (FBG), 2-h postprandial glucose (2 hPG), and glycated hemoglobin (HbA1c) levels compared to those carrying male fetuses; however, these differences were not statistically significant ($P > 0.05$). Furthermore, we observed that mothers with male fetuses exhibited higher triglycerides (TG) and total cholesterol (TC) levels and a greater predisposition towards diabetes relative to mothers with female fetuses ($P < 0.05$). These observations contrast with previous findings from a singleton pregnancy cohort study ($n = 299$), which demonstrated that after adjusting for several demographic and clinical factors related to MetS, a female fetus was significantly more associated with maternal insulin resistance than a male fetus ($P = 0.001$) (*Xiao et al., 2014*). A previous retrospective study including 617 singleton child women (mean age, 32.4 ± 4.9 years) suggested that after adjustment for various diabetes-related factors, insulin resistance was significantly higher and insulin sensitivity was lower in the mothers with female fetus group than in the mothers with male fetus group (*Yamashita et al., 2020*). Young women with gestational diabetes have a high risk of developing type 2 diabetes for life. Moreover, B-cell dysfunction underlies the initial development of gestational diabetes but also later progression to diabetes in these high-risk patients (*Feig et al., 2008*). In a study with an average follow-up of 5.5 years, *Retnakaran & Shah (2016)* found that women carrying female fetuses had a higher risk of developing diabetes postpartum (adjusted hazard ratio = 1.06, 95% CI [1.01–1.12]). Considering this observed greater risk of rapidly developing diabetes after giving birth, it is possible that women who develop gestational diabetes during pregnancy with a female fetus may be more at risk of impaired B-cell function compared to those with gestational diabetes while with a male fetus (*Retnakaran et al., 2015*; *Retnakaran & Shah, 2015*).

In contrast, the present study shows that the prevalence of MetS in mothers carrying male fetuses was higher than in those carrying female fetuses among postmenopausal women ($P < 0.05$). These data are consistent with those from an earlier systematic review and meta-analysis that reported an association between male fetal sex and term preeclampsia as well as gestational diabetes. This meta-analysis suggested that pregnancy complications may depend on fetal sex, where maternal cardiovascular and metabolic load may be greater when carrying a male fetus (*Broere-Brown et al., 2020*). Another study of 1,074 pregnant women indicated that a male fetus could be associated with worse β-cell function, higher postprandial glucose levels, and a greater risk of gestational diabetes in the mother ($P < 0.05$) (*Retnakaran et al., 2015*). Similar findings were obtained by comparing 55,891 mothers carrying a male fetus and 53,104 carrying a female fetus, showing a statistically significant association between male fetuses and higher rates of gestational diabetes (*Sheiner, 2007*). However, gestational diabetes could lead to later problems for the mother, such as recurrence, increased risk of developing type 2 diabetes, MetS, and CVD

(*Bekedam et al., 2002*). Possible causes include an elevation in metabolic activity that could result in greater vulnerability of the male fetus during crucial developmental stages (*Poon et al., 2019*). Another important reason is that a male fetus could unfavorably affect the mother's β-cell compensatory response (*Retnakaran et al., 2015*). How this may occur is not yet clear, but one possible aspect of the male fetus (where the Y chromosome could be involved) could affect the placenta's production of hormones or proteins involved in making up for B-cell deficiency (*Retnakaran et al., 2015*).

In our study, we also found the prevalence of MetS was statistically greater in the mothers with male fetus group than those with female fetus group among postmenopausal women. But the mechanism by which fetal sex difference influences MetS especially in postmenopausal women remains unexplained. Firstly, menopause nearly adversely affects all components of MetS (*Pu et al., 2017*). This is because, after menopause, the secretion of reproductive hormones in women decreases (including estrogen, progesterone, and testosterone) (*Lee et al., 2022*). Estrogen plays a regulatory role in various aspects of glucose and lipid metabolism as well as adipokine secretion. The deficiency of estrogen leads to an increase in obesity and abdominal obesity, hyperlipidemia, impaired glucose metabolism, and insulin resistance (*Kim, Cho & Kim, 2014*). Secondly, some authors have reported that the sex of the fetus may interact with certain polymorphisms in the mother, such as progesterone receptor (*Hocher et al., 2009*), angiotensin converting enzyme (*Hocher et al., 2011*), and peroxisome proliferator-activated receptor gamma2 (*Kousta et al., 2020*), which could have an effect on the pregnant mother's glucose regulation. A recent study shows that women carrying a male fetus exhibit a more proinflammatory (G-CSF, IL-12p70, IL-21, and IL-33)/proangiogenic immune milieu (PlGF and VEGF-A) than women carrying a female fetus (*Enninga et al., 2015*). Inflammation is an important contributor to the pathophysiology of MetS (*Neeland et al., 2024*). Thirdly, in China, mothers are more concerned about the care and anxious of their boys. Girls were found to be better at filial caring of their parents and maintaining basic living standards for the elderly (*Du & Ding, 2004*). For this high-risk population, we recommend regular screenings for metabolic syndrome. Early detection through routine check-ups can facilitate timely interventions. Additionally, we advocate for significant lifestyle modifications, including the adoption of a balanced diet rich in calcium and vitamin D and engaging in moderate physical activities like walking or swimming.

## STRENGTHS AND LIMITATIONS

The strength of this study lies in its pioneering exploration of the relationship between fetal sex and metabolic syndrome in women over 40 years old within a Chinese cohort. However, the study has several limitations. First, there was no information available regarding maternal obesity and diabetes during pregnancy. Second, we did not analyze the relationship between fetal sex and the components of MetS. Third, as this study is cross-sectional, the exact mechanism between fetal sex and MetS remains unclear. Finally, our study population was derived from Luzhou, China, so the findings may not be generalizable to other populations worldwide.

In summary, fetal sex was not associated with MetS in Chinese women aged 40 and older. However, among postmenopausal women, mothers carrying male fetuses showed a significantly higher frequency of MetS. More attention should be paid to postmenopausal women who have carried male fetuses, and early measures should be taken to prevent related chronic diseases.

## ABBREVIATIONS

| | |
|---|---|
| **MetS** | metabolic syndrome |
| **TG** | triglycerides |
| **HDL-C** | high-density lipoprotein cholesterol |
| **LDL-C** | low-density lipoprotein cholesterol |

## ACKNOWLEDGEMENTS

We are thankful to all the participants in this study.

### Funding

This work is supported by the grants 2017YFC1309805-1 and 2023YFS0078 from the Ministry of Science and Technology. The funders had no role in study design, data collection and analysis, decision to publish, or preparation of the manuscript.

### Grant Disclosures

The following grant information was disclosed by the authors:
Ministry of Science and Technology: 2017YFC1309805-1 and 2023YFS0078.

### Competing Interests

The authors declare that they have no competing interests.

### Author Contributions

- Qian Xie conceived and designed the experiments, performed the experiments, analyzed the data, prepared figures and/or tables, authored or reviewed drafts of the article, and approved the final draft.
- Ruoqing Li performed the experiments, analyzed the data, prepared figures and/or tables, and approved the final draft.
- Qin Wan performed the experiments, authored or reviewed drafts of the article, and approved the final draft.
- Nanwei Tong conceived and designed the experiments, performed the experiments, authored or reviewed drafts of the article, and approved the final draft.

### Human Ethics

The following information was supplied relating to ethical approvals (*i.e.*, approving body and any reference numbers):

The Ruijing hospital approved this survey under number 52/2014.

## Data Availability

Raw data are provided as a Supplemental File.

## Supplemental Information

Supplemental information for this article can be found online at http://dx.doi.org/10.7717/peerj.19380#supplemental-information.

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
