# Peer review of "Association between fetal sex and metabolic syndrome in women aged 40 years and older: the REACTION study"

_PeerJ, doi:10.7717/peerj.19380_

## Round 0.1 · original submission · Major Revisions

You must fully address the comments of the reviewers. This includes clarifying / correcting the age and sex criteria for the participants

Reviewer 1 ·

Basic reporting

This is an interesting study.
Lot of recent literature in the field have been used.
However, language needs to be improved.
Most of the time two study groups are referred to as male group and female group. Reword them as mothers with male fetus and female fetus to sound more scientific

Experimental design

This is not original research; however, it is interesting to explore this topic in different populations.
Authors are exploring the possibility of linking sex of the fetus with MS. MS is attributed to many factors. Therefore, if the authors can include a more comprehensive justification on the topic, it will add more scientific value
Study population is from a longitudinal study evaluating the risk of cancer in people with diabetes. The study population already have one out of three criterias to classify as MS. There is a bias with this.
Glucose tolerance test is done with 82.5 g glucose, and 75 g is the dose given in guidelines. why?
please include inclusion and exclusion criteria clearly

Validity of the findings

Discussion can be improved by including the mechanisms for various theories given and to support findings
Line 205-206 : "we should advocate equality between...... to get rid of son preference"". Why is this statement necessary for this manuscript? Nothing related to this is discussed previously in the document.
Results indicate postmenopausal women who had a male fetus are at more risk and should take early measures. It is better to mention what are the measures authors are thinking and how to execute those as this is a common statement in most of the manuscripts and nothing really happens after the paper is published.

Additional comments

Nothing

Reviewer 2 ·

Basic reporting

I received for review an original research article entitled "Association between fetal sex and metabolic syndrome in women 40 and older: the REACTION study", prepared by Qian Xie, which was submitted to the PeerJ. Metabolic syndrome and its components are a very important issue for public health in the modern world. Metabolic syndrome and its components, especially type 2 diabetes, significantly increase the risk of developing cardiovascular diseases and the risk of cardiovascular events. Cardiovascular diseases are the leading cause of morbidity and mortality in the modern world. Conducting research that can contribute to a better understanding of metabolic syndrome is therefore very important. I appreciate the efforts made by the authors of the manuscript. However, I would like to draw attention to the need for significant corrections to the submitted manuscript and for it to be subjected to a second review.
1) It is necessary to describe more precisely from which study the collected data comes, as well as who actually participated in the study. The description so far is not fully understandable to me. On the one hand, I learned that the study involved pregnant people over 40 years of age, on the other hand, one of the exclusion criteria is age over 85. This needs to be sorted out, because the current description is unclear to me.
2) From the presentation of the results we learn that men also participated in the study. The title of the study suggests that the study was about pregnant women over the age of 40. This cannot be. This needs to be organized and described in a precise way. The current title is completely inadequate.
3) The description of the statistical analysis is very sparse and should be supplemented. How was the conformity of the distribution of a given variable to the normal distribution tested? Was the mean and standard deviation used for variables with a normal distribution? Was the median and interquartile range used for variables with a non-normal distribution? What specific tests were used to test statistical hypotheses? How was correlation tested?
4) The strengths and limitations of the study should be described precisely.
5) It is good that the authors indicated the criteria for diagnosing metabolic syndrome, and thus its most important components. However, it is worth expanding on the description of metabolic syndrome (in the introduction or discussion) and showing that metabolic syndrome also shows a relationship with dysfunction of other organs. I propose to mention the recently published results of studies that suggest that subclinical fluctuations in thyroid function have an impact on the diagnosis of metabolic syndrome and body composition analysis parameters (10.3390/medicina60071080), and this translates into some parameters related to the function of the cardiovascular system (10.3390/medicina60091445).

Experimental design

No further comments.

Validity of the findings

No further comments.

Additional comments

No further comments.

---

## Round 0.2 · Minor Revisions

Some remaining minor revisions are still needed, as detailed by Reviewer 1 (to double check the entire manuscript for spelling and grammer)

Reviewer 1 ·

Basic reporting

Authors still need to go through the article carefully and attend to spelling and grammar mistakes

Experimental design

fine

Validity of the findings

fine

Reviewer 2 ·

Basic reporting

I received for review a revised version of the original research article entitled "Association between fetal sex and metabolic syndrome in women 40 and older: the REACTION study", prepared by Qian Xie, which was submitted to the PeerJ. In my opinion, the manuscript has been significantly improved. The authors have satisfactorily addressed my concerns expressed in the first review. I believe that in its current version the manuscript presents satisfactory substantive and cognitive value. I have no further critical remarks. I recommend the manuscript for publication in its current form.

Experimental design

No further comments.

Validity of the findings

No further comments.

Additional comments

No further comments.

---

## Round 0.3 · accepted · Accept

Congratulations!
Prof. Yoshinori Marunaka, M.D., Ph.D.